# Characterization of Polyurethane Foam Waste for Reuse in Eco-Efficient Building Materials

**DOI:** 10.3390/polym11020359

**Published:** 2019-02-19

**Authors:** Raúl Gómez-Rojo, Lourdes Alameda, Ángel Rodríguez, Verónica Calderón, Sara Gutiérrez-González

**Affiliations:** Departamento de Construcciones Arquitectónicas e I.C.T., Escuela Politécnica Superior, C/Villadiego S/N 09001, University of Burgos, 09001 Burgos, Spain; rgrojo@ubu.es (R.G.-R.); lalameda@ubu.es (L.A.); arsaizmc@ubu.es (Á.R.); vcalderon@ubu.es (V.C.)

**Keywords:** polymer waste, polyurethane foam, leaching test, microstructure

## Abstract

In the European Union, the demand for polyurethane is continually growing. In 2017, the estimated value of polyurethane production was 700,400 Tn, of which 27.3% is taken to landfill, which causes an environmental problem. In this paper, the behaviour of various polyurethane foams from the waste of different types of industries will be analyzed with the aim of assessing their potential use in construction materials. To achieve this, the wastes were chemically tested by means of CHNS, TGA, and leaching tests. They were tested microstructurally by means of SEM. The processing parameters of the waste was calculated after identifying its granulometry and its physical properties i.e., density and water absorption capacity. In addition, the possibility of incorporating these wastes in plaster matrices was studied by determining their rendering in an operational context, finding out their mechanical resistance to flexion and compression at seven days, their reaction to fire as well as their weight per unit of area, and their thermal behaviour. The results show that in all cases, the waste is inert and does not undergo leaching. The generation process of the waste determines the foam’s microstructure in addition to its physical-chemical properties, which directly affect building materials in which they are included, thus offering different ways in which they can be applied.

## 1. Introduction

According to the latest report published by Plastic Europe-the Facts 2017 [1], the demand for plastic in Europe in 2016 was 49.9 MTn, 3.1% higher as regards to 2014. Of this demand, 7.5% is polyurethane, which implied an annual demand of 3.78 MTn in 2017. Of the 3.78 MTn, approximately 70% is in the form of foam (1.40 MTn of flexible foam, 1.22 MTn rigid foam), 30% being that of polyurethane elastomers and other products. Of the 2.62 MTn of PU foam, approximately 27% of waste is generated (700,400 Tn), of which, 31.1% is recycled (220,000 Tn), 41.3% is incinerated (294,278 Tn) and the remaining 27.3% is taken to landfill (193,120 Tn). The sectors according to demand are: Construction and building (24.5%); automotive (19.5%), refrigeration (21.3%) and other sectors within the textile industry, usage in technology, etc. (34.7%). The majority of polyurethane products such as low- and high-density foam are thermostable [2]. The material is characterized by its lattice structure, maintaining its shape and resistance under high-pressure and high-temperature conditions that end up degrading, thus after it has been manufactured, and after it reaches gelation point, the material cannot be melted in order to be remodeled into other products. As a consequence of this, the recycling process of thermostable polyurethanes is complex and unprofitable (chemical, mechanical and thermochemical recycling) [3]. As regards recovery techniques based on incineration in order to regain energy, there are environmental disadvantages due to the emission of atmospheric contaminants such as HCB dioxins and the emission of fine particles [4]. These disadvantages, along with the large amount of waste taken by landfills, prompt a search for alternative ways to recover this type of waste. Over the past decade, European obligations to control the environmental impact of waste incineration (Directive 2000/76/CE) [5] and of landfill of waste (Directive 2008/98/CE) [6] have led to the increased cost of these waste treatment options. These costs will increase as more strict controls are introduced; as taxes on landfill and on incineration increase, this further encourages reuse.

Several studies have researched the option of reusing polyurethane foam waste, combined with pitch binders, and PU foam waste as a dry aggregate in different cement or gypsum matrices. Studies on cement and PU mortars have shown that there is a positive influence of these recycled aggregates on their manufacturing, which ensures excellent durability, even with regard to other traditional aggregates [7]. Previous research has led us to think that this polymer is able to reduce the amount of sand in cement mortars by substituting sand with PU by between 13–33% [8], 25–50% [9] or even 25–100% [10], all of these accounting for substitution in volume. The choice of the volume of substitution depends on the characteristics that are desired to be achieved in the final product. Products that are considerably more flexible and hydrophobic than other conventional materials [11] could potentially be obtained. In reference to research on gypsum material with polyurethane, results establish the compatibility of PU waste with a gypsum-based aggregate by combining different amounts of the PU waste in order to obtain a new cladding material for façades with thermal insulation properties [12]. Other research has advanced the design of this material, incorporating it in prefabricated gypsum materials that is extremely lightweight and has thermal and sound insulation properties. Laboratory tests have been carried out to improve the mechanical properties of these materials by means of the inclusions of additives and fibres [13]. Nevertheless, there is still a long way to go in order to optimise the properties of these materials, fundamentally, in terms of their fire reaction properties and acoustic improvements. One of the key parameters for this is a thorough study of the waste’s physical, chemical and micro structural characteristics, which can vary depending on its provenance.

The aim of this research is centred on the analysis of the properties of five polyurethane wastes from different industries with a view to assess their potential reuse in prefabricated, gypsum-based construction materials. The intention is to provide an alternative to the current practice of incineration and recycling options in accordance with the criteria established in the European Parliament Directive 2008/98/CE and the European Council 19 November 2008 Directive on waste.

## 2. Materials and Methods

In order to determine the viability of using polyurethane waste cells and outline the possibility of its use in new building materials, five types of wastes were selected from different industries and chemical characterization tests were carried out using elemental analysis (CHNS), thermal gravimetric analysis and waste leaching test. In order for the waste to be incorporated into new materials as a dry aggregate, it must be previously processed. The granulometry is then determined and the processing parameters are calculated, in addition to a physical characterization analysis, which determines the real and apparent density, the ability to absorb water. Finally, the wastes are microstructurally characterized using Scanning Electron Microscopy (SEM).

### 2.1. Materials

Five polyurethane foams from different industries were analysed (Figure 1).
(P): Rigid yellow polyurethane foam waste, in powder, compressed into pellet form (pellets). The waste is generated in the manufacture of insulation panels for the refrigeration sector, at Paneles Aislantes Peninsulares (PAP) factory in Cuenca, Spain. The waste is produced by trimming edges during the production stage.(B): Rigid yellow polyurethane foam waste, in the form of plates (Block). Waste generated in the manufacture of insulation panels for the refrigeration sector at Paneles Aislantes Peninsulares (PAP) factory in Cuenca, Spain. The waste comes from rejected panels and remnants of panels used in factory tests.(I): Rigid yellow polyurethane foam waste in the form of plates (block). The waste is generated in the manufacture of insulation panels for the refrigeration sector and comes from factory waste, from Italpannelli factory in Zaragoza, Spain.(A): Semi-rigid grey polyurethane foam waste, which comes in pieces and powder form; it is compressed into a pellet shape. The waste is generated in the manufacture of automobiles at Grupo Antolín IGA factory in Beaumont, France.(SG): Semi-rigid polyurethane foam waste; they are remains of car seats from scrapped cars obtained from the company Sigrauto in Madrid, Spain.

### 2.2. Methodology

#### 2.2.1. Elemental Analysis (CHNS)

This technique is used for the quantitative determination of carbon (C), hydrogen (H), nitrogen (N), and sulphur (S) in all sample types, to obtain the oxide content, measured as a percentage of the weight. The equipment used is a LECO Analyzer CHNS-932 and VTF-900. The analysis technique is fully automated, and is based on the combustion of the samples under optimum conditions (*T* = 950–1100 °C in pure oxygen atmosphere) to convert the aforementioned elements into simple gases (CO_2_, N_2_, H_2_O and SO_2_) to achieve a quantitative determination of C, N, H and S content.

#### 2.2.2. Waste Leaching Test

This test was carried out according to the UNE-EN 12457-2 Standard [14]. A sample of waste is placed in a bottle with a certain amount of water, and placed in a stirring device for 24 h. Once the mixture is stirred and filtered, the eluate (liquid to be tested) is obtained. Tests such as pH, electrical conductivity (EC), salt and TDS (total dissolved solids) were carried out. It is necessary to make a blank with distilled water, before testing the eluate.

#### 2.2.3. Thermogravimetric Analysis (TGA)

This technique measures the change in mass of a sample, while being heated at a constant speed. In this case, it is used to find out the degradation temperature of the waste, thus outlining the working temperature of the material. The waste samples used in this test were previously processed. The equipment used is a Q600 thermal analyser TA Instruments (TGA/DSC), which simultaneously provides a true measurement of the same sample from room temperature to 1500 °C of heat flow (DSC) and weight change (TGA). It has a dual balance mechanism, a twin conductor, and horizontal purge gas system with mass flow control and gas switching capability. This equipment is joined via a TG interface to an F-TIR spectrometer, which also facilitates simultaneous analysis by infrared spectroscopy of the gases produced in the decomposition of the substances studied.

#### 2.2.4. Density

To calculate the real density of the polymer waste test principles for natural stone, the UNE-EN 1936: 2007 Standard [15] were applied, using the pycnometer method. It is necessary to crush a sample of raw material until a fineness of particle capable of passing through a 0.063 mm sieve is achieved. A 10 g sample of material in isopropyl alcohol is placed into the pycnometer and then weighed. The pycnometer is cleaned, filled with isopropyl alcohol again, and reweighed. The real density (in kg/m³) is calculated by means of the ratio between the mass of the dry and crushed test piece, and the volume of liquid displaced by the mass.

The apparent density of the wastes used in this research was calculated in the exact condition that they are in when they are received from where the waste is generated (as a slab and in pellet form) and once it has been transformed by means of being processed, the relationship between weight/volume is calculated. In the case of non-processed waste, 1 kg of waste is taken and the volume that this occupies is calculated. In the case of processed waste, the procedure consists of filling a container with a specific volume and it is weighed, then the weight/volume equation is applied. In this case, a 1 L capacity container was used as a reference, which was filled with crushed waste and then weighed.

#### 2.2.5. Water Absorption Capacity

This test applies standard UNE-EN 13755:2008 [16]; it consists of placing the material to be tested (dry and with constant mass) into a container filled with water until fully covered, for 24 h. At the end of that period of time, the material is weighed and placed back into the water for another 24 h. The material is weighed yet again; if constant weight that does not differ from the previous day is observed, the material has reached saturation.

The following equation is then applied:

Ab = (Saturated weight − Dry weight)/Dry weight) × 10088(1)

#### 2.2.6. Laser Granulometry

The different foams were crushed and their granulometric size determined through laser granulometry diffraction using a HELOS 12K SYMPATEC analyser. The samples were analysed for 15 s in an isopropyl alcohol suspension.

#### 2.2.7. Processing Parameters

The processing parameters were defined by determining cutting time, crushing time, and the energy of the crushing.

It is necessary to process the polyurethane foam to be used in the tests. (B), (SG) and (I) wastes are split into smaller pieces. These pieces are placed in a RETSCH SM100 Mill, where they undergo a crushing and sieving process (Figure 2). Pellets (P and A) are directly placed into the crusher.

#### 2.2.8. Scanning Electron Microscopy (SEM)

This technique allows for the microscopic structure of the different polyurethanes (closed-cell open-cell) to be discovered. For this test, the waste samples did not undergo processing.

The equipment used is a Microscope FEI Quanta-600, which allows for the samples to be observed and characterized by obtaining high-resolution imaging of organic and inorganic materials at high magnifications. The equipment can be used in high vacuum, acting as a traditional scanning electron microscope (SEM), and can work in environmental mode (ESEM). The latter mode allows observation without coating or metallizing the sample, which makes it a non-destructive technique. The equipment is also used alongside two sets of X-ray microanalysers, the EDX and WDX Oxford, which allows for elemental analysis in a timely manner, or the compositional mapping of specific areas of the materials studied.

#### 2.2.9. Mechanical Properties of the Gypsum/PU Mixtures

The following mechanical properties of the mixtures were tested at 28 days: Flexural and compressive strength. These tests were carried out in accordance with standard EN 13279-2 [17]. Flexural strength is determined by the load needed to break a prism-shaped specimen measuring 160 × 40 × 40 mm^3^, which is lain on rollers positioned at 100 mm intervals. The test was performed on a minimum of three specimens. Compressive strength tests are performed on the broken sections of the specimens after they have been previously tested to flexural failure (at least six samples underwent compressive strength tests). The test consists of applying a load to a 40 × 40 mm^2^ section of the sample.

#### 2.2.10. Thermal Properties of the Gypsum-PU Mixtures

Thermal conductivity was measured according to standard EN 12667 [18]. The guarded hot plate and heat flow meter methods were used to carry out this test. These methods involve establishing a constant relationship and uniformity in the ratio between the heat flow density on the inside of the homogeneous samples measuring 300 × 300 × 30 mm^3^ and those of a set of plane parallel faces. The specimens were dried to a constant mass at a temperature of 35 °C and analyzed in a Laser Comp heat flow meter.

The non-combustibility test was carried out to evaluate the behaviour of the samples at high temperatures. This test was carried out in accordance with standard EN ISO 1182 [19]. The tests were conducted in an open vertical furnace, in which a cylindrical specimen with a diameter of 75 mm and a height of 150 mm was placed. During the test, the electronically controlled furnace temperature was increased at a constant rate, from room temperature to 800 °C during a period of 2 h, after which, the temperature remained at 800 °C for a further 60 min. In this test, increases in temperature were measured by the furnace thermocouple. The duration of flaming and the mass loss of the sample was calculated, which had previously been conditioned in a ventilated oven at temperatures of (60 ± 5) °C, over 24 h.

The gross heat of combustion (gross calorific value) test, which is carried out according to standard EN ISO 1716 [20], determines the maximum potential heat released by a product when it reaches complete combustion. In this test, a specimen of a minimum mass of 50 g is burned at constant volume in an oxygen atmosphere in a bomb calorimeter by benzoic acid combustion. The specific combustion heat under these conditions is calculated on the basis of the observed increase in temperature, taking into account the loss of heat and the latent heat of water vaporization.

## 3. Results and Discussion

### 3.1. Elementary Analysis (CNHS)

Table 1 shows the results of the carbon, nitrogen and sulphur components of each of the wastes that were analysed. As was expected, carbon was the majority component of all the polymer wastes. In each case, they had a similar percentage of carbon. The analysis also confirmed the existence of hydrogen and nitrogen in smaller proportions with respect to the waste as a whole. As regards the other components that each waste has, it could be asserted that as this is a case of polyurethane, there is a significant amount of oxygen. Similar observations have been noticed in [21] and other components associated with the possible impurities each foam may contain. For example, as regards foams that come from scrapped vehicle seats (SG), it can contain metals linked to elements from the actual seat such as copper or aluminium, which will later be identified in the scanning electron microscopy test. On no occasion was the presence of sulphur detected.

### 3.2. Thermogravimetric Analysis (TGA)

The TGA test results show the % loss of weight of the different wastes when the temperature increases.

Wastes (P) and (B) come from the same company and have the same isocyanate polyol component composition. The difference between them is the presence of metal impurities that (P) has with respect to (B), which shows as being totally clean. This difference is noted in the loss of mass, which in the case of (P) occurs at 280 °C, and that does not occur in waste (B). In both cases, the first degradation occurs at around 200 °C, polymer decomposition occurs from 325 °C to 550 °C (Figure 3).

In the case of foam (I) (Figure 4), it shows a very similar behavior to that of foam (B). Both of them come from the insulation for refrigeration industry and have a similar initial mass loss at 238° C and the total decomposition of the polymer occurs between 345 °C–450 °C.

In the case of flexible foam (SG), a minimal loss of mass was observed at 280 °C, probably due to the metal impurities that car seats contain. The loss of mass corresponding to the polymer’s decomposition occurs between 400 °C and 550 °C (Figure 5a). In the case of foam (A), three different mass losses occur (Figure 5b). The first of these losses occurs at 320 °C with a significant degradation of the material. The second loss occurs at 400 °C, which corresponds to the polymers’ degradation and the last stage occurs at 500 °C, which corresponds to the loss of other components in this foam.

Similar effects have been noticed in [22], in which the urethane bond groups of PUR start to break up into isocyanates segments and polyols segments from about 200 °C with a second loss in the temperature range of 350–500 °C. These results indicate that the thermal behaviour of the material is acceptable. Thus, this thermal analysis technique is a highly useful tool for studying the reuse of these polymers, with no chemical or physical changes detected.

### 3.3. Scanning Electron Microscopy (SEM)

Generally speaking, polymeric cellular materials can be defined by a two-phased structure in which the gaseous phase, stemming from a foaming agent, whether physical or chemical, was dispersed throughout a solid polymeric matrix [23]. Foam is a specific type of cellular material that is generated by the expansion of a material in liquid form. This is the case of wastes type (B), (I) and (SG) that were analysed in this research. However, there are processes subsequent to foaming that cause a loss of cellular structure of the polymer. This is the case for wastes (P) and (A) that are obtained by means of a milling process that generates particles of an extremely fine nature, which is also compressed. This causes a structure with layers of polymer that are very different from that of the wastes obtained differently and have sheets of foamed polyurethane (Figure 6a–d).

The cellular materials are classified according to their cellular structure and cell connectivity. In the case of foam type (B), the structure is an intermediate structure and it can be seen that a portion of the cellular structures is formed by an open-cell structure, while the other portion of the cellular structure is formed of a closed-cell structure (Figure 7a,b). In this case, the walls are of a 10 µm thickness and the cells have a diameter of between 10 µm and 200 µm. Foam type (I) has a closed-cell structure in which the gas is occluded in the interior of the cells. The cells are largely homogeneous in terms of the size of which they are comprised, between 50–400 µm (Figure 7c,d). Flexible foam (SG) has an open-cell structure, where gas can freely circulate between the cells since they are interconnected with each other, which can cause an improvement in acoustic properties. Other authors have verified this effect in works on acoustic damping performance in flexible polyurethane foams [24] (Figure 7e,f). In this case, a characteristic that is typical of this type of foam can be observed, that is the presence of pores in the cells’ interconnecting walls, as well as the presence of metal impurities.

### 3.4. Waste Leaching Test

One of the processes that must be monitored when using waste materials as raw materials in construction is leaching. It is common for the materials to be in contact with water or dampness, which could cause a leaching process [25]. In view of the results obtained from the leaching test (Table 2), it can be observed that the electrical conductivity does not exceed the maximum value permitted (3000 µs/cm). As for the maximum amount of the total of dissolved solids (500 mg/L), on no occasion was this amount exceeded. The values were always lower than 100 mg/L. The pH levels were also within the permitted range (5.5–9). However, authors [26] obtained contrary results that are related with toxicity in polyurethanes (artificial leather, floor coating and children’s handbag), which showed that hydrophobic compounds were causing the toxicity. The polyurethanes waste studied in this work do not show hydrophobic behavior. It can, therefore, be established that the wastes analyzed do not display any contaminating behavior when in contact with water or dampness, thus they can be used in construction materials.

With regard to the anti-aging characterization of PU foam waste, previous studies related to durability testing in PU waste have been carried out wherein PU forms a part of the matrix of a cement mortar construction material [27]. The water in the 105 °C test analyses the degree of resistance when the material is exposed to boiling water and consists of accelerated ageing under high humidity conditions. The dry heat at 140 °C test consists of ageing under dry heat at 140 °C for 240 h [28]. Thermal oxidation of the polyurethane could occur under these conditions, resulting in variations in molecular weight that are evident by the reduction of mechanical strength and by a change in colour. Intense oxidation, especially at high temperatures, could lead to the deterioration of the polymeric chain and the loss of carbon monoxide and water. After carrying out both tests, it was observed that the presence of PU decreases the percentage of expansion and decreases the possibility of there being alkaline reactivity, which in the cases of mortars, means an increase in structural stability over time. As a consequence, the conclusion has been reached that PU does not degrade after the tests indicated and, therefore, does not impair the end behaviour of the construction material.

### 3.5. Processing Parameters

In [Fig polymers-11-00359-ch001], the preparation (cutting) and crushing times are outlined, as well as the energy used in the processing of 1 kg of waste.

In this paper, wastes in slab form (I), board (B) as a whole (SG) and in pellet form (P), (A) were studied. In the case of wastes (I), (B) and (SG), it was necessary to cut them prior to placing them into the shredder. It must be taken into consideration that the process is carried out on a laboratory scale in which the shredder has a limited input capacity. In this case, the waste that needed the least amount of cutting time was waste (I), explained by the fact that it has a compact closed-cell structure and is more rigid than the other foams. The foam that needed the most amount of time was type (SG), 60 min shredding per kg of the sample, as this is a flexible, highly pliable, and difficult-to-handle foam. As regards machine shredding time, the values proportionally vary to the prior cutting time. In waste type (P), the duration of shredding took the least amount of time (6 min). This is followed by type (I) with 20 min. The longest time is for waste (SG); there was additional difficulty in working with the waste due to the machine’s sieve becoming blocked due to the nature of this type of foam. 

### 3.6. Laser Diffraction Granulometry

The granulometric study focussed on sizes less than 1 mm. [Fig polymers-11-00359-ch002] shows the granulometric results of the different wastes in sizes smaller than 1 mm. In view of the results, it can be observed that wastes in powder form (P), (A) have a very similar particle size with an average diameter of 229 µm and 271 µm, respectively. The waste in the form of a slab (I) has a diameter of 194 µm, noticeably greater than the board shaped waste (B). The most significant difference can be observed in waste (SG) with an average particle size of 401 µm and parts that can reach 772 µm. The particle size distribution in the different wastes, alongside the real density values, will determine the final mechanical properties of the construction material [29]. Such is the case of plasters with dry aggregate of waste polyurethane that will be studied in the following section.

### 3.7. Determination of Apparent and Real Density, Water Absorption Capacity

Another parameter that strongly determines the material’s final properties and as a consequence how it can be applied is density. 

It is necessary to find out the density of the material prior to being processed and after processing in order to determine the conversion efficiency of the waste. On looking at the results of Table 3, the following can be observed. The highest densities of the polyurethane prior to being processed are found in wastes that are compressed into pellet form (P), (A), which indicate that in a lesser volume, there is space for a larger amount of the material. This factor is due to the material’s extreme fineness and to the polyurethane being arranged in layers. This was previously observed microstructurally by SEM imagery. The polymer with the lowest apparent density was the flexible waste (SG) with 93% less compared with pellet. As can be observed, the apparent densities are, in cases where the waste is not compressed, very low. This indicates that there was a problem related with the storage of this type of waste as well as its transportation and keeping, since a low weight of the materials occupies a large volume [30]. 

In light of the apparent density results after processing the waste, it can be observed that the lowest apparent density pertains to the wastes with a cellular structure (B), (I) and (SG). This may be due to the granulometry of these polyurethanes which, once processed, have a lesser variety of sizes with an increased percentage of specific sizes ([Fig polymers-11-00359-ch002]). As expected, the apparent density of the wasted that is in a compacted form, (P) and (A), increases after being processed, as this destroys its cellular structure and reduces its pores on account of the cells and as a consequence, this increases its density.

The water absorption capacity of each waste varies according to the foam’s structure and morphology [31]. Thus, the flexible foam (SG) is the foam that showed the greatest absorption capacity. This is probably due to the highly porous nature of its cells, which is further accentuated by the presence of pores in between the cell walls. As it is a case of an open-cell structure (Figure 7e,f), the water enters the interior of the foam more easily. Both waste (B) and waste (I) showed lower absorption capacity caused by a semi-closed and closed-cell structure, respectively.

### 3.8. The Possibilities of the Uses of the Wastes Studied

One of the ultimate aims of this research is to improve the use of these types of foams and expanding this use in the industries that generate polyurethane, thus improving the ratio of the volume of reused PU. One option is to incorporate the processed waste as a dry aggregate in plaster matrices. Different substitutions of plaster type A1 for rigid PU foam waste [32,33] have been made in previous studies. The conclusion was reached that the optimal ratio of components in volume could be (1/1.5), that is, 1 part of gypsum with 1.5 parts of polyurethane foam waste. This study used all of the wastes characterized in this paper.

The following test results will now be shown: Mechanical resistance to compression and flexion at 28 days (Table 4) and fire reaction by means of non-combustion test and the gross heat of combustion test (Table 5). Of the samples that are shown, the best behaviour, the weight per unit of area and thermal conductivity, was calculated compared with a standard plaster.

In all cases, the mechanical resistance obtained met the requirements outlined in the regulations with over 2 MPa of resistance to compression and over 1 MP of flexion resistance. The wastes with the smallest particle sizes give the best results in the mechanical properties of the mixtures. Samples 1/1.5 (B) and 1/1.5 (I) have a similar granulometry and similar results in mechanical properties. If these two are compared with the result of sample 1/1.5 (P), which has a larger average particle size, it can be observed how the values obtained are considerably lower [34]. As regards the fire reaction properties, initially, the samples underwent the non-combustion test. The thermal behaviour of the samples, confirmed by the non-combustibility test, gives us an idea of their fire retardance properties. The results of the non-combustibility test (Table 5) confirmed that the samples that included polyurethane in their composition, and specifically, the 1/1.5 (B) sample and 1/1.5 (I), did not have flaming times of less than 20 s. The samples had a temperature increase of below 50 °C and losses of less than 50% of their mass. This result indicated that even if the contribution of the materials to fire reaction is taken into account, their composition corresponded to Euroclass A1 (non-combustible), in accordance with the European fire reaction classification of building materials for homogeneous products [35]. In order to check this classification, the Superior Calorific Power was calculated showing a value of below 2 MJ/kg. Therefore, these materials can be classified as non-combustible. The rest of the mixtures with other wastes did not meet the minimum standards required in the non-combustible test. They will be need to be tested with regulation EN-13823 and EN ISO 11925-2 in order to check classifications A2 or lower. It was noted that in the SEM tests, these wastes had impurities due to metal contamination or adhesives, which would be the reason why they did not reach the minimal requirements established in order to be classified as A1.

Mixtures 1/1.5 (B) sample and 1/1.5 (I) were tested in order to determine their weight per unit of surface and their thermal conductivity, which are two important properties when it comes to determining their characteristics when in use. The values from both tests are shown in Table 6.

In view of the results, it can be observed that the materials composed of Gypsum-PU-(B) had a 30% and 33% reduction in surface weight with respect to the standard plaster, which is explained by a lower real density of waste type (B) 1370.9 kg/m^3^ and 1105.0 kg/m^3^ of waste (I) with respect to that of the plaster that it substituted, which is 2650 kg/m^3^. Given that the thermal conductivity depends on density and on the characteristics of the actual PU waste [36], the values were reduced by up to 36% with regard to mixtures 1/1.5 (I) with respect to the conventional plaster, which gives rise to an improvement in the material’s thermal insulation.

## 4. Conclusions

Five PU wastes from different sectors and industries were chosen in order for there to be a wider scope for PU to be reused, and for it to be easier for the project to be replicated allowing polyurethane waste to begin to be used in different sectors.

All of the polymers degrade at above 200 °C. In the case of polyurethane SG, degradation occurs at a higher temperature (400 °C).None of the PU wastes have a leaching capacity and they are all considered to be suitable for use in new construction materials.The wastes that had been compacted had the best processing times, with the same prior cutting time and low energy use. This characteristic that these types of foams display, along with the fact that they have the greatest apparent density, create an advantage with respect to the other wastes with regard to PU being productively reused in building materials, both in terms of transportation (from the factory where the waste is generated) and in the collection of the waste and the rendering of the mixture.It was observed that the polyurethane that underwent a milling process had a high level of fineness with average particle sizes being around 250 µm and had greater levels of apparent density in respect of the rest of the wastes. The flexible foam had a larger average particle size of approximately 400 µm. In this case, the apparent density is lower compared to the rest of the foams.The microstructure of the polyurethanes is different depending on the industry from where they came. In the case of board and slab shaped wastes that come from the refrigeration industry, the structure is hexagonal semi-closed celled. Open and closed cells can be observed in the images from the SEM. The waste from the refrigeration industry is in slab form (I) and has a closed-cell structure. In both cases, adequate thermal behaviour was predicted, which could be used in improving thermal insulation when they are included in construction material.It is observed that SG waste has a structure suited to be used as a possible acoustic absorber, because of its open pore structure.The wastes that result from milling processes have a structure that is in the form of overlapping layers with no defined hexagonal structure. In this case, the wastes had some metal impurities in their structure associated with the actual milling process.As a final aim of this research, there is the possibility of including the wastes in plaster matrices in ratio with volume (1/1.5) thus obtaining adequate mechanical resistance to compression of over 2 MPa, a reduction in thermal conductivity by 33%, and a reduction in the weight of the material by 31%. In regards to the non-combustibility test and calorific value test, only the rigid PU foam wastes (B), and (I) met the standards to have an A1 classification, which is ideal for interior cladding materials for buildings. There was worse fire reaction behaviour in samples (P), (A) and (SG) due to the impurities that they contain. Nevertheless, it must be determined whether the classification in these two cases would be that of A2 or worse and alternative ways for the material to be applied in different areas of a building will need to be found.

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
