# Peer review of "Characterization of Polyurethane Foam Waste for Reuse in Eco-Efficient Building Materials"

_polymers, 2019, doi:10.3390/polym11020359_

Round 1

Reviewer 1 Report

This manuscript addresses a novel issue as it characterizes polyurethane wastes -obtained from different industries- and studies their potential to be used as alternative materials for the production of building products. In particular this article explores the viability of incorporating five different polyurethane wastes in a gypsum matrix. This research is important as it will help to produce ecofriendly building materials, and thus helps minimizing the environmental impact caused by the construction activity.

It is an interesting paper, clear and concise. Thus, I consider the topic and research presented has the quality to be published in the journal.

The following minor change is suggested:

· In section 2, I suggest presenting first the definition of each polymeric foam waste used, in order to understand the abbreviations, and then indicate the type of industry.

·  Please revise the number of the sections and subsections. Section 3.1 should be 2.1, etc.

·  Please change “Kg” to “kg” in the text and tables.

·  I think that the results shown in section 3.8 (tables 4 -6) were taken from the previous research works. In order to clarify, I suggest adding the references at the end of the title of the tables.

Author Response

The authors are grateful to reviewer for the valuable comments. We attempted to respond to the reviewer’s comments correctly and clearly in this response letter. Items corrected are highlight in yellow in the text.

This manuscript addresses a novel issue as it characterizes polyurethane wastes -obtained from different industries- and studies their potential to be used as alternative materials for the production of building products. In particular this article explores the viability of incorporating five different polyurethane wastes in a gypsum matrix. This research is important as it will help to produce ecofriendly building materials, and thus helps minimizing the environmental impact caused by the construction activity.

It is an interesting paper, clear and concise. Thus, I consider the topic and research presented has the quality to be published in the journal.

The following minor change is suggested:

1)     In section 2, I suggest presenting first the definition of each polymeric foam waste used, in order to understand the abbreviations, and then indicate the type of industry.

In line with the reviewer’s suggestions, the order of the definition of each waste in the text has been modified by the authors.

2)     Please revise the number of the sections and subsections. Section 3.1 should be 2.1, etc.

This mistake has been corrected. Thank you.

3)      Please change “Kg” to “kg” in the text and tables.

This mistake has been corrected. Thank you very much.

4)      I think that the results shown in section 3.8 (tables 4 -6) were taken from the previous research works. In order to clarify, I suggest adding the references at the end of the title of the tables.

The results shown in section 3.8. didn’t take from previous research works. That which has been taken from previous research is the choice of dosage (1/1.5). The references to the research paper in which the previous research works has been verified as being between the range of (1/1-1/2), are [31] and [32]. This is the reason because the authors have included the methodology followed in this study in order to obtain the mixtures of plaster with polyurethane mechanical resistance, thermal conductivity and reaction to fire results.  The following information has been added to the experimental section (please see the text added to the experimental section of the document):

“Mechanical properties of the gypsum/PU mixtures:

The mechanical properties of the blends (flexural and compressive strength) was measured at 28 days according to standard EN 13279-2 [17]. Flexural strength is determined by the load needed to break (160 × 40 × 40 mm3) prism shaped specimens supported on rollers positioned at 100 mm intervals. The test was performed on a minimum of three specimens. Compressive strength tests are performed on the broken sections of the specimens previously tested to flexural failure (on at least six samples). The load is applied to a (40 × 40 mm2) contact section.

Thermal properties of the gypsum-PU mixtures:

Thermal conductivity was measured according to standard EN 12667 [18], using the guarded hot plate and heat flow meter methods. These methods involve establishing a constant relation and uniformity in the ratio between heat flow density in the interior of the homogeneous samples (300 × 300 × 30 mm3) and a set of plane parallel faces. The specimens were dried to a constant mass at a temperature of 35 °C and analysed in a Laser Comp heat flow meter.

The non-combustibility test was carried out to evaluate the behavior of the samples at high temperatures, in accordance with standard EN ISO 1182 [19]. The tests were conducted in an open vertical furnace, in which the specimen was placed within a cylindrical space with a diameter of 75 mm and a height of 150 mm. During the test, the electronically controlled furnace temperature was increased at a constant rate, from room temperature to 800 ºC in 2 h, after which, it remained at 800 ºC for a further 60 minutes. In this test, temperature increases were measured by the furnace thermocouple, the duration of flaming and the mass loss of the sample was calculated, in cylindrical samples of 45 mm in diameter, and 50 mm in height, which had previously been conditioned in a ventilated oven at temperatures of (60 ± 5) ºC, over 24 hours.

The gross heat of combustion test (calorific value) according to regulation EN ISO 1716 [20] determines the maximum potential heat given off by a product when it is fully burned.  In this test a specimen of minimum 50g mass is burned at constant volume in an atmosphere of oxygen in a bomb calorimeter by benzoic acid combustion. The specific combustion heat under these conditions is calculated on the basis of the observed increase in temperature, taking into account the loss of heat and the latent heat of water vaporization.”

Reviewer 2 Report

The article analysis polyurethane foam waste and gypsum-polyurethane-based composites. The article is of poor quality, there are no methodologies for determination of compressive and flexural strengths as well as thermal conductivity, the text structure need to be proofread by a native English speaker, there are no discussion with other authors work. In my opinion, the work presented into highly impacted journal should be much better; therefore, I suggest not accepting the article.

Author Response

The authors are grateful to reviewer for the valuable comments. We attempted to respond to the reviewer’s comments correctly and clearly in this response letter. Items corrected are highlight in yellow in the text.

Comments and Suggestions for Authors:

The article analysis polyurethane foam waste and gypsum-polyurethane-based composites. The article is of poor quality, there are no methodologies for determination of compressive and flexural strengths as well as thermal conductivity, the text structure need to be proofread by a native English speaker, there are no discussion with other authors work. In my opinion, the work presented into highly impacted journal should be much better; therefore, I suggest not accepting the article.

We are grateful for the reviewer’s comments. In the “Methods” section, we have included the methodology followed in the compression and flexion tests, as well as that of the thermal conductivity and reaction to fire tests.

The English structure of the text has been reviewed and improved by a native English speaker. Enclosed is a professional translator’s certification, which endorses the quality of the English editing in the article.

The discussion part of the results has been improved including more references to authors in the text ([26],[30],[33]).

Reviewer 3 Report

In this work, the behavior of various waste polyurethane foams was analyzed. The approach and results are sufficiently described, but the explanation from the experiment results in the paper is from a scientific point of view not sufficiently. Here is some minor comments:

The authors better to give an anti-aging characterization of PU

Most of the Figures are not clear. For example, the text in figure is too small to be seen, please enlarged them.

Is there some relationship between the particle size distribution of waste and the mechanical properties of building materials?

Suggest cite a related reference: Macromolecular Rapid Communications 2018, 39, 1800635

Author Response

The authors are grateful to reviewer for the valuable comments. We attempted to respond to the reviewer’s comments correctly and clearly in this response letter. Items corrected are highlight in yellow in the text.

In this work, the behavior of various waste polyurethane foams was analyzed. The approach and results are sufficiently described, but the explanation from the experiment results in the paper is from a scientific point of view not sufficiently. Here is some minor comments:

1)     The authors better to give an anti-aging characterization of PU

In line with the reviewer’s recommendation, in the section on the characterization of raw materials, the authors have commented on aspects related with the durability of PU when forming part of the matrix of a construction material (see paragraph in lines 287-299 of the text).  In light of this, a reference has been added (see [26]) which includes the results on this polymer’s durability as per standard EN ISO 2440.

Paragraph in lines 287-299 of the text:

With regard to anti-aging characterization of PU foam waste, previous studies about durability test in PU waste have carried out when forming part of the matrix of a construction material of cement mortar [26]. Water at 105 ºC test analyses the degree of resistance when the material is exposed to boiling water and consists of accelerated ageing under high humidity conditions. The dry heat at 140ºC test consists of ageing under dry heat at 140ºC for 240 h [27]. Thermal oxidation of the polyurethane could occur under these conditions, obtaining the result of variations in molecular weight that are evident by the reduction of mechanical strength and by a change in color. Intense oxidation, especially at high temperatures, could lead to deterioration of the polymeric chain and the loss of carbon monoxide and water. After carrying out both tests, it was observed that the presence of PU decreases the percentage of expansion and decreases the possibility of there being alkaline reactivity, which in the cases of mortars means an increase in structural stability over time. As a consequence, it is concluded that PU does not degrade after the tests indicated and therefore does not impair the end behaviour of the construction material.

2)     Most of the Figures are not clear. For example, the text in figure is too small to be seen, please enlarged them.

The review is right. With the aim of improving this, the authors have increased the size of the text in figures 3a, 3b, 4, 5a and 5b

3)     Is there some relationship between the particle size distribution of waste and the mechanical properties of building materials?

The authors have included an explanation of the relation between the distribution of the particle size of the waste and the mechanical properties of the construction material. Please see the new paragraph included in the text: The smallest of the waste’s particle size gives the best results in the mechanical properties of the mixtures. Samples 1/1.5(B) and 1/1.5(I) have similar granulometry and similar results in mechanical properties. If these two are compared with the result of sample 1/1.5 (P) with a bigger average particle size, it can be observed how the values obtained are considerably reduced [33]”.

4)     Suggest cite a related reference: Macromolecular Rapid Communications 2018, 39, 1800635

In line with the reviewer’s recommendation, this new bibliography reference has been added. Please see [30].

Round 2

Reviewer 2 Report

Authors have addressed almost all of my comments, however, the references introduction into the text does not provide any clarity of discussion part. Please improve your results and discussion part with comparison with other similar scientific works, i.e. "...similar observations have been noticed in [x]", "...however, authors [x] obtained contrary results which are...", etc.

Author Response

The authors thank the reviewer for their comments. The authors have attempted to improve the discussion of the results by comparing these discussions to those of similar scientific papers cited in the text.

Thank you very much for consider this manuscript for publication as a research paper in Polymers.

Sincerely yours,

Sara Gutiérrez González

Round 3

Reviewer 2 Report

Thank you for your trust. I am really interested to review this article again. Authors improved their article very much but the formatting of this article is really poor. Overall, it is nice to see that there was really much additional work done to improve this work. Again, please format your article according to the requirements of polymer MDPI template.